# Effect of optical correction by fully corrected glasses on postural stability

**Ji In Bae, Dong-Sik Yu, Sang-Yeob Kim** *

Department of Optometry, College of Health Science, Kangwon National University, Samcheok-si, Republic of Korea

* syk@kangwon.ac.kr

**Data Availability Statement:** All relevant data are within the manuscript and its Supporting Information files.

**Funding:** The author(s) received no specific funding for this work.

## Abstract

Although various previous studies have reported that the experimentally induced refractive errors interfered with postural control, few studies have demonstrated the optical correction effect of wearing glasses. The purpose of this study was to investigate whether wearing full corrected glasses to correct myopia and hyperopia can have a positive effect on postural stability. To this end, a total of 34 subjects (19 males and 15 females) of an average age of 22.38 ± 2.41-years-old participated in this study. After measuring the full corrected powers of refractive errors of subjects through subjective refraction, updated glasses were provided to 17 myopic subjects and first time glasses were provided to 17 hyperopic subjects as full corrected glasses, respectively. Postural evaluation was carried out using the TETRAX biofeedback system, after which we compared and analyzed the postural instability index and sway power index before and after wearing full corrected glasses. When updated and old glasses for correcting myopia were worn, the postural instability index was significantly reduced, and the sway power index was statistically decreased only in the mid-high frequency region associated with the somatic system, compared to the no glasses state, respectively. However, after wearing first time glasses for hyperopia correction, no significant difference was found in the postural instability index or sway power index. We suggest that providing optimal visual information through the optical correction of myopic refractive error is a useful approach that can lead to synergistic effects of somatic functions involved in postural control. Consequently, we demonstrated that wearing glasses to fully correct the refractive errors has a positive effect on increasing postural control in static posture. Our results may have important clinical implications in the field of optometry and balance evaluation.

## Introduction

For postural stability, sensory information received from the somatic system, vestibular system and visual system must be appropriately controlled with the motor nervous system through sensory integration in the central nervous system [1,2]. The somatic system is involved in postural control by recognizing the position, movement and balance of the body's musculoskeletal system in space [3]. The vestibular system is related to the rotational movement of the head and mainly functions to provide information on the body's position with respect to gravity and movement [4]. As age increases, these sensory organs lead to changes in muscle structure

**Competing interests:** The authors have declared that no competing interests exist.

and sensory function, which in turn reduce overall postural control [5,6]. In addition, damage to the vestibular organs including the inner ear, damage to the peripheral nerves due to degenerative diseases and diabetes, and damage to the central nervous system due to various brain lesions are well known to be pathological causes that interfere with postural control [7–9].

Among the sensory organs contributing to postural control, the visual system plays an essential role in stable postural control by continuously providing information on the body's position with respect to the environment by first recognizing objects [10]. Paulus [11] and Lord et al. [12] found that 20~70% more body sway was observed when both eyes were completely blinded and visual information was completely blocked. Further, many previous studies have reported that postural control is easily reduced alongside various visual problems. Vijay et al. [13] reported that postural instability was significantly increased in a cataract simulation group as compared to the non-simulation group in the cataract simulation study. Hortense et al. [14] reported that macular degeneration is an important visual disorder associated with postural instability, and Aachal et al. [15] reported that visual defects also increase the risk of falling in patient with glaucoma. Low vision patients with severe visual handicaps have difficulties even in daily life, including with leisure activities, due to their reduced postural control and limited mobility, severely affecting their quality of life [16]. In addition, poor gaze, which is characteristic of people with strabismus and vergence dysfunction, is also known to be one of the factors causing postural instability [17,18]. The findings of these studies suggest the importance of examining the impact of visual information on postural control.

Refractive error refers to a condition in which parallel rays entering the eye fail to focus on the retina, such as in the case of myopia which focuses in the front of the retina, hyperopia which focuses on the back of the retina and astigmatism which forms a non-focus astigmatism [19]. With regard to previous studies on the relationship between refractive error and postural stability, Edwards [20] reported that postural instability increased by more than 50% when 50 subjects participated in myopia with a spherical lens of +5.00 D. Paulus et al. [21] reported similar findings when myopia was induced with spherical lenses of +4.00 D and +6.00 D; postural instability is increased by about 25% compared to previously. Among other studies, Paulus et al. [22] assessed posture in naked eyes with myopia wearing corrective eye glasses from -3.00 D ~ -11.00 D. As a result, it was found that 25% more body sway, on average, occurred compared to when wearing corrective glasses. It is clear that the blurred visual information caused by myopia is a visual factor that hinders postural stability. As mentioned above, although the types of refractive errors vary, most of the preceding studies are limited to conditions that cause myopic blur. In our own previously conducted studies, we analyzed the effects of refractive errors on hyperopia, astigmatism and inequality, as well as on postural stability and the risk of falling [23, 24]. However, the results of our previous studies were obtained by experimentally incurring refractive errors using lenses which may be limited in their ability to determine the optical correction effect linked to wearing glasses. Therefore, this study sought to demonstrated whether myopes and hyperopes would experience a positive effect on static postural control from optical correction by wearing full corrected glasses (updated glasses for myopes or first time glasses for hyperopes) compared to not wearing glasses. Furthermore, we identified the cause using the Fourier transformation analysis of sway power index provided by the TETRAX system, in which postural stability is increased by each set of full corrected glasses.

## Materials and methods

### Subjects

This study included 34 subjects (19 males and 15 females) of a mean age of 22.39 ± 2.30 years-old. Among the 34 subjects, 17 subjects, of an average age of 21.18±1.59 years, had myopic

refractive errors, including simple myopia and myopic astigmatism, and 17 subjects of an average age of 23.59±2.53 years had hyperopic refractive errors, including simple hyperopia and hyperopic astigmatism. Among all subjects, 17 myopes were already using prescription glasses (old glasses) and 17 hyperopes were not using prescription glasses. All subjects were physically healthy, and verbal questioning confirmed that there were no muscle diseases, systemic diseases, eye diseases and medications related to body balance or falling. In addition, subjects who had signs and symptoms related to vergence dysfunction or a corrected binocular visual acuity under 0.9 were excluded from this study. This study was approved by the Kangwon National University Institutional Review Board, and conducted in accordance with the tenets of the Declaration of Helsinki.

## Instruments and procedures

In this study, postural evaluation was performed using the TETRAX® static posturography system (Tetrax Potable Multiple System, Tetrax Ltd., Ramat Gan, Israel) (Fig 1). The TETRAX biofeedback system is a device designed to assess the overall balance of the body, in which four

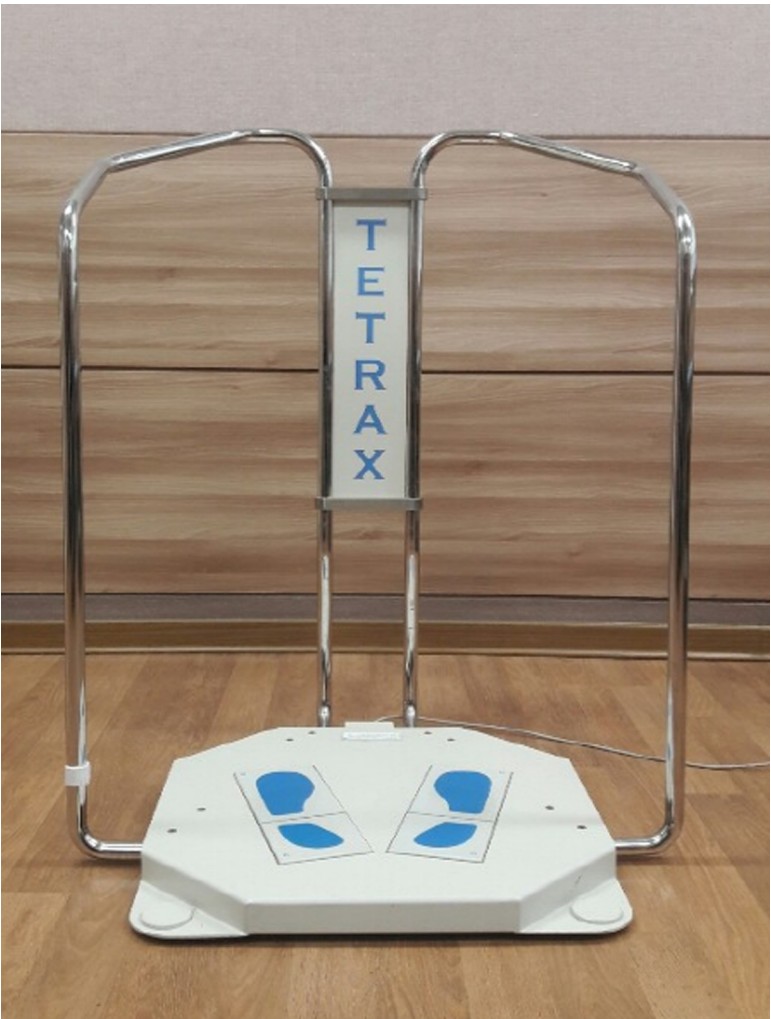

**Fig 1. TETRAX® static posturography system used in this study.**

force plates marked A (left heel), B (left toe), C (right heel) and D (right toe) are installed to measure pressure changes in static posture (Fig 2).

The measurement is performed for 32 seconds according to the system's guidelines. At the end of the measurement, the information output from the four force plates is converted into digital signals, and the postural instability index and sway power index by the Fourier transform can be analyzed by comprehensively analyzing area, length and velocity of the sway and movement of the gravity center of the posture.

The postural instability index is calculated based on the concept that the higher the stability, the less the change in the weight on the four force plates. Since the index represents the overall postural instability by measuring the degree of posture sway on the four force plates, the larger the posture instability index value, the more often or the higher the % change in the weight of the four force plates. Therefore, it may be estimated that the higher this index value, the more unstable the posture [25]. The postural instability index also increases with age as a result of degraded postural control (11.69±2.21 in normal young subjects and 24.84±6.07 in old subjects over 65 years old) [26]. A Fourier analysis is a mathematical representation of the wavelength signals of the body's vibrations in a horizontal plane created by a patient to maintain a vertical posture. The vibration intensity in each region is calculated by subdividing various frequency components included in the measured value when the body sway occurs on the force plates through a Fourier transformation of postural sway [27]. The sway power index is divided into four frequency domains, as follows, and the causes of the increased body sway can be analyzed for each sensory organ. At first, the low-frequency region is in the range of 0.01–0.1 Hz, which means that an abnormally increased value in this region is associated with visual

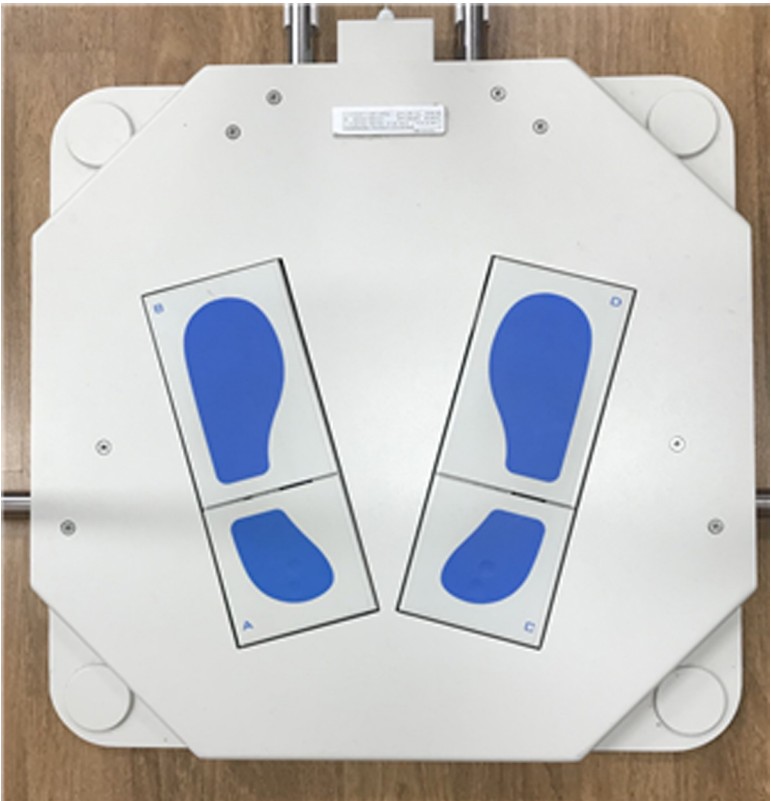

**Fig 2. Four plates on the TETRAX® static posturography device.**

dysfunction. Secondly, the low-medium frequency region is in the range of 0.1–0.5 Hz, indicating that abnormally increased values in this region are associated with peripheral vestibular disturbances. Thirdly, the medium-high frequency region is in the range of 0.5–0.75 Hz, which means that the abnormal increase in this region is associated with somatic dysfunction. Finally, the high-frequency region is in the range of 1.0–3.0 Hz, which means that an abnormal increase in this region is associated with disorders of the central nervous system.

To evaluate the optical correction effect of refractive error on postural stability, the subjective refraction test was first performed using a manual phoroptor (Ultramatic RX Master, Reichert, USA) and a 6-m LCD visual acuity table (LUCID'LC, Everview, Korea), and the final correction refractive powers of each subject was detected based on a MPMVA (Maximum Plus to Maximum Visual Acuity) method for full correction [28]. After ordering the spectacle lenses corresponding to the final corrective refractive powers of all 34 subjects, full corrected glasses were provided by accurately matching each subject's PD (pupillary distance) and OH (optical height). The measurement conditions for posture evaluation were set to the no glasses state, the old glasses state (only for myopic subjects), and the full corrected glasses state (updated glasses for 17 myopes or first-time glasses for 17 hyperopes). The measurement of each condition was carried out randomly for each subject. For posture assessment, each subject was asked to align their feet on the four force plates of TETRAX with their shoes removed and to assume an anatomical posture. After having settled for 10 seconds in that state, the measurement was carried out for 32 seconds according to the measurement manual. During the posture evaluation, each subject was asked to look at a point fixed 6 m ahead in order to minimize changes in the eye lens's accommodation reflex. Based on the measured data, the changes in the postural instability index and the sway power index were compared and analyzed under each test conditions.

## Data analysis

Data analysis was performed using the SPSS program (Ver. 21 for window, SPSS Inc., Chicago, IL, USA). The paired t-test and repeated-measures analysis of variance (repeated-measures ANOVA) methods were used to analyze the changes in postural control in each measurement condition. For all analyses, it was estimated that there was a statistically significant difference when $p < 0.05$.

## Results

Table 1 summarizes the glasses prescriptions for 17 myopic subjects. The average equivalent spherical power of their old glasses was S-3.92 ± 1.82 D, and the average equivalent spherical power of full corrected prescriptions for updated glasses was S-4.11 ± 1.92 D. Table 2 presents the full corrected prescriptions in hyperopes. In 17 hyperopic subjects, the average equivalent spherical power of full corrected prescriptions for their first time glasses was S+0.43 ± 0.29 D.

After wearing the corrective glasses in the 17 myopes, the average change in postural instability index is shown in Fig 3. The postural instability index was significantly decreased when wearing myopia correction glasses as compared to the value measured in the no glasses state (F = 4.561, $p < 0.05$ by repeated measures ANOVA). A post-hoc analysis revealed that the postural instability index was significantly decreased in the old glasses state and after the wearing the updated glasses compared to the values measured in the no glasses state ($p < 0.05$ for no glasses vs. old glasses, $p < 0.05$ for no glasses vs. updated glasses by LSD post hoc analysis). However, compared to the old glasses, the postural instability tended to decrease after wearing the updated glasses, but there was no statistically significant difference ($p > 0.05$ for old glasses vs. updated glasses by LSD post hoc analysis).

**Table 1. Information of individual full corrected prescription in myopic subjects.**

| No | Full corrected-prescription (for updated glasses) | | Current-prescription (for old glasses) | |
|---|---|---|---|---|
| | RE | LE | RE | LE |
| 1 | S-1.75 | S-0.75 C-0.75 Ax5 | S-2.25 | S-1.50 C-0.50 Ax180 |
| 2 | C-1.75 Ax90 | S-0.50 C-1.25 Ax90 | S-0.50 C-1.00 Ax90 | S-0.75 C-1.25 Ax90 |
| 3 | S-4.25 | S-3.75 C-0.50 Ax75 | S-4.50 C-0.25 Ax15 | S-4.25 |
| 4 | S-4.25 C-1.25 Ax10 | S-4.50 C-1.25 Ax180 | S-4.25 C-1.50 Ax10 | S-5.00 C-1.25 Ax180 |
| 5 | S-2.25 | S-3.50 C-0.50 Ax180 | S-3.25 | S-4.25 C-0.50 Ax160 |
| 6 | S-5.50 C-0.50 Ax5 | S-5.50 C-0.25 Ax175 | S-5.75 C-0.75 Ax180 | S-5.50 C-1.00 Ax180 |
| 7 | S-3.50 C-2.00 Ax175 | S-3.50 C-1.75 Ax180 | S-4.25 C-2.75 Ax180 | S-4.25 C-2.50 Ax180 |
| 8 | S-3.50 C-2.00 Ax5 | S-3.50 C-2.00 Ax180 | S-3.00 C-1.75 Ax180 | S-3.00 C-1.25 Ax180 |
| 9 | S-7.50 C-0.75 Ax10 | S-6.25 C-1.25 Ax140 | S-7.00 | S-6.50 |
| 10 | S-4.25 C-1.25 Ax5 | S-4.75 C-1.50 Ax175 | S-4.00 C-1.00 Ax180 | S-4.50 C-1.00 Ax180 |
| 11 | S-5.25 C-0.50 Ax180 | S-5.25 C-0.75 Ax180 | S-4.25 C-0.50 Ax180 | S-4.25 C-0.75 Ax175 |
| 12 | S-0.25 C-1.75 Ax175 | S-1.50 C-0.50 Ax 5 | C-1.50 Ax180 | S-1.00 C-0.50 Ax180 |
| 13 | S-4.00 C-2.00 Ax175 | S-5.25 C-2.00 Ax170 | S-3.75 C-1.50 Ax175 | S-3.75 C-1.50 Ax160 |
| 14 | S-3.50 C-0.50 Ax20 | S-3.25 C-0.75 Ax150 | S-3.25 C-0.25 Ax50 | S-3.00 C-0.50 Ax145 |
| 15 | S-3.00 C-0.75 Ax5 | S-3.00 C-1.75 Ax175 | S-2.50 | S-2.25 |
| 16 | S-6.50 | S-6.75 C-0.50 Ax180 | S-5.75 | S-6.25 |
| 17 | S-1.75 | S-1.25 | S-1.25 | S-1.00 |

RE: Right eye, LE: Left eye, S: Spherical lens power (Diopter), C: Cylindrical lens power (Diopter), Ax; Axis (˚).

Fig 4 shows the analysis results of comparing the average postural instability of the 17 hyperopes with first-time glasses. After wearing hyperopia corrective glasses, the postural instability

**Table 2. Information of individual full corrected prescription in hyperopic subjects.**

| No | Full corrected-prescription (for first-time glasses) | |
|---|---|---|
| | RE | LE |
| 1 | S+0.50 C-0.50 Ax180 | S+0.75 C-0.75 Ax180 |
| 2 | S+0.75 C-0.25 Ax175 | S+2.00 C-1.00 Ax15 |
| 3 | S+0.75 C-0.50 Ax90 | S+0.25 C-0.25 Ax90 |
| 4 | S+0.50 | S+0.75 C-0.50 Ax130 |
| 5 | S+0.75 | S+1.00 |
| 6 | S+0.50 C-0.25 Ax105 | S+0.75 C-0.50 Ax95 |
| 7 | S+1.00 C-0.75 Ax70 | S+0.75 C-0.25 Ax120 |
| 8 | S+0.75 C-0.50 Ax90 | S+0.50 C-0.50 Ax90 |
| 9 | S+0.50 C-1.00 Ax90 | S+0.50 C-0.50 Ax95 |
| 10 | S+0.50 C-0.25 Ax175 | S+0.50 C-0.25 Ax175 |
| 11 | S+0.25 C-0.25 Ax90 | S+0.50 |
| 12 | S+1.00 C-0.75 Ax15 | S+0.50 C-0.25Ax15 |
| 13 | S+0.75 C-1.25 Ax75 | S+0.50 C-0.75 Ax105 |
| 14 | S+1.00 C-0.50 Ax160 | S+0.75 C-0.25 Ax180 |
| 15 | S+1.00 C-2.00 Ax105 | S+0.50 C-1.75 Ax90 |
| 16 | S+0.25 C-0.25 Ax175 | S+0.75 C-0.50 Ax170 |
| 17 | S+0.75 C-0.50 Ax35 | S+0.50 |

RE: Right eye, LE: Left eye, S: Spherical lens power (Diopter), C: Cylindrical lens power (Diopter), Ax; Axis (˚).

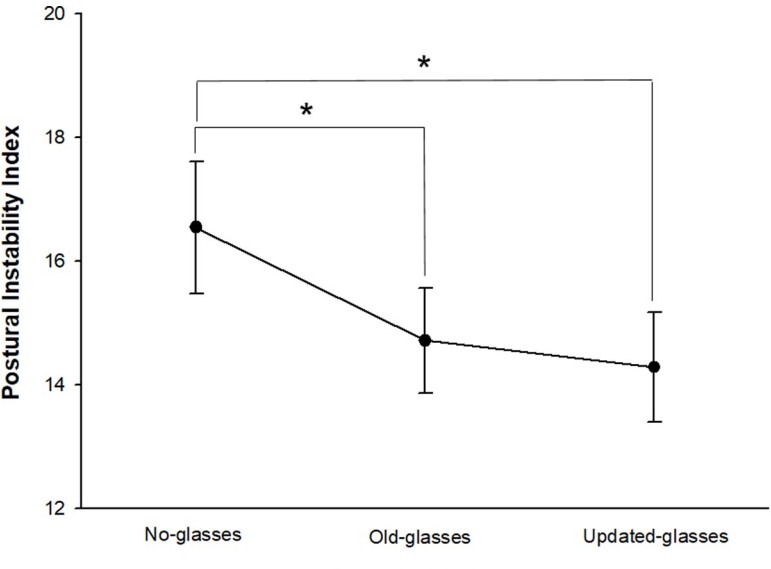

**Fig 3. Changes in the postural instability index depending on visual conditions in myopic subjects.** $^{*}p < 0.05$: significantly different depending on each visual condition according to LSD (least significant difference) post hoc analysis by repeated measures ANOVA. Error bars indicate the standard error (SE) of the mean.

index was more decreased than the no glasses state, but there was no statistically significant difference (t = 1.006, $p > 0.05$ by paired t-test).

Table 3 shows the results of having analyzed and compared the sway power index in the four frequency regions after wearing the corrective glasses in 17 myopic subjects. A significant difference in sway power index was only observed in the mid-high frequency region among 4 frequency regions, depending on visual conditions (F = 4.724, $p < 0.05$ by repeated measures

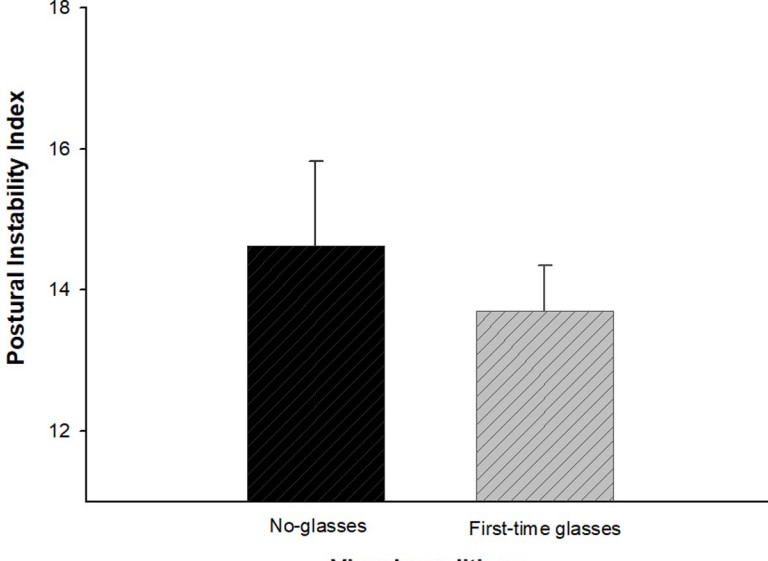

**Fig 4. Changes in the postural instability index depending on visual conditions in hyperopic subjects.** Error bars indicate the standard error (SE) of the mean.

**Table 3.** Changes of sway power index in each frequency range depending on visual conditions in myopic subjects.

| Visual conditions | Sway power index in 4 frequency ranges | | | | N |
|---|---|---|---|---|---|
| | low | low-medium | medium-high | high | |
| no glasses | 26.52±3.01 | 8.17±0.59 | 3.35±0.21[a] | 0.58±0.04 | 17 |
| old glasses | 22.02±3.14 | 8.37±0.70 | 2.86±0.17[b] | 0.52±0.04 | 17 |
| updated glasses | 26.05±2.43 | 8.49±0.54 | 2.81±0.18[c] | 0.58±0.05 | 17 |
| F/p-value | 0.938/0.393 | 0.088/0.916 | 4.724/0.021* | 0.928/0.386 | |
| Post-hoc | - | - | a>b, a>c | - | |

Data are expressed as mean±SE.

*$p < 0.05$: significantly different depending on each visual condition according to repeated measures ANOVA.

[a,b,c]: subgroups by LSD (least significant difference) post-hoc analysis.

ANOVA). A post-hoc analysis revealed that the sway power index in the mid-high frequency was significantly decreased in the old glasses state and after wearing the updated glasses compared to the values measured in the no glasses state ($p < 0.05$ for no glasses vs. old glasses, $p < 0.05$ for no glasses vs. updated glasses by LSD post hoc analysis). However, compared to the old glasses, postural instability tended to decrease after wearing the updated glasses, but there was no statistically significant difference ($p > 0.05$ for old glasses vs. updated glasses by LSD post hoc analysis). The changes in the sway power index in each frequency region before and after wearing the first-time glasses in 17 hyperopic subjects are shown in Table 4. Contrary to myopes, there was no significant difference in sway power index in all frequency regions even though fully corrected glasses were worn.

## Discussion

Uncorrected refractive errors easily reduce visual acuity, contrast sensitivity and stereoscopic function, which are visual factors which contribute to postural stability [29,30]. Many previous studies [20–22,31–33] have reported that, experimentally, the induced refractive errors reduce postural stability, but it is hard to find a study documenting the effects of optical correction by wearing full corrected glasses. Therefore, the purpose of this study was to generate useful data from specialty fields related to optometry and balance evaluation by analyzing the effects of full optical correction on posture control in 17 myopes and 17 hyperopes who needed corrections in the form of updated glasses or by getting glasses for the first time.

### Effects of full optical correction for myopes on postural stability

As shown in Fig 3, after 17 myopic subjects had worn updated glasses, the postural instability index was significantly reduced compared to the no glasses state, and a similar result was

**Table 4.** Changes of sway power index in each frequency range depending on visual conditions in hyperopic subjects.

| Visual conditions | Sway power index in 4 frequency ranges | | | | N |
|---|---|---|---|---|---|
| | low | low-medium | medium-high | high | |
| no glasses | 21.19±9.36 | 6.89±2.62 | 3.01±1.19 | 0.54±0.18 | 17 |
| first-time glasses | 20.88±8.37 | 6.92±1.92 | 2.82±0.80 | 0.53±0.17 | 17 |
| t/p-value | 0.146/0.903 | -0.608/0.944 | 2.450/0.425 | 0.126/0.760 | |

Data are expressed as mean±SE.

found when wearing their old glasses compared to no glasses. Therefore, our findings confirmed that the optical correction of refractive errors by wearing glasses actually has a positive effect on postural stability. Lord et al. [30,34] reported that the decrease in depth perception and contrast sensitivity was a major visual factor increasing the risk of falling by reducing postural stability. Nevitt et al. [35] reported that poor stereoacuity was a crucial visual factor increasing the risk of falling. According to the results from this study and other previous studies [20–22,31–33], we reaffirmed that myopic blurring acts as a factor in increasing postural instability by compromising visual functions. Therefore, we emphasize that the optical correction of myopic refractive errors is essential to providing proper visual information that contributes to stable postural control. In addition, as shown in Fig 3, postural instability tended to decrease slightly when wearing the updated glasses with an average equivalent spherical power of -4.11 ± 1.92 D compared to old glasses with an average equivalent spherical power of -3.92 ± 1.83 D. We suggest that full correction for myopic blurring, even though it has a low residual power of S-0.25 D, may provide the optimal visual input for improving postural control. We further suggest that the effect of full optical correction should be considered clinically meaningful in optometric clinical contexts.

## Effects of full optical correction for hyperopes on postural stability

Hyperopia is defined as the refraction error in which the parallel rays entering into the eye are focused behind the retina. Unlike myopia, hyperopic individuals may have good vision, without any optical correction through the accommodation of the eye lens [19]. Recently, we reported another study yielding an interesting result in which postural stability was reduced despite a mean visual acuity of 1.0 or higher through an accommodation mechanism, in the context of hyperopia induced by a spherical lens of –1.00 D [36]. In this study, first time glasses for correcting hyperopia were worn by 17 subjects. As a result, the postural instability index decreased slightly, but there was no statistically significant change (Fig 4). We believe this is due to the fact is that the average equivalent spherical power of the hyperopic subjects in this study was S+0.43 ± 0.29 D, close to emmetropia conditions. As mentioned above, hyperopia tends to receive less attention than myopia because of its ability to maintain good vision, even if it is not optically corrected. Therefore, further analysis is required with regard to the effect on the optical correction for an uncorrected hyperopia of 1.00D or more. We also expect that the optical correction of hyperopia is another clinically important factor in postural control.

## Fourier transformation analysis of sway power index before and after wearing corrective glasses

In order to assess the cause of the increased postural stability by optical correction of refractive errors, the variation of sway power index was analyzed by a Fourier transform method provided by the TETRAX system. Tables 3 and 4 show the results of the comparison and analyses of the changes in sway power index before and after wearing corrective glasses in myopic and hyperopic subjects, respectively. Related clinical studies using the TETRAX system demonstrate the usefulness of the Fourier analysis. The excessive sway in a particular frequency region is explained either by the presence of a pathological problem in the sensory organ by compensatory efforts [37,38]. Taguchi [39] and Kollmitzer et al. [40] reported that in patients with peripheral vestibular pathologies, the sway power index was only increased in the low-medium frequency regions. Further, it was confirmed that the increase in the sway power index in the medium-high frequency region was a sign of somatic dysfunction related to lower limb, spine and back motion [40]. According to a clinical study by DeWit [37], increased sway

in the high-frequency region is often a symptom of central nervous system symptoms associated with tremor, which is interpreted as an abnormality in the cerebellum, cerebral cortex or proprioception. In our previous studies, we reported that induced hyperopia increased the sway power index in the low-medium frequency region. We explained that an excess accommodative reflex for focusing on the retina caused the autonomic imbalance to affect the vestibular system associated with postural control [36]. However, in this study, it was shown to reduce the sway power of the medium-high frequency region after optical correction, a characteristic which was only obvious in myopic subjects after wearing old glasses and updated glasses, respectively, compared to the no glasses state (Table 3). Therefore, we expect that by removing the blurry image by the optical correction of myopia and providing an optimal visual information to the sensory integration area responsible for posture control, the effect of the somatic system among sensory organs can be stabilized. This seems to be factor contributing to increasing overall postural control. However, the results of this study are limited since they were based on the results collected over the course of 32 seconds after wearing the full corrected glasses, whether in the form of updated or first-time glasses. Therefore, further research should be conducted to further analyze the adaptation phenomenon when wearing glasses, as well as the time-dependent changes following optical correction.

In summary, static postural stability was improved when glasses were worn to fully correct each type of uncorrected refractive errors. Clinically, the optical effect was more remarkable in the myopes than in the hyperopes when comparing with a no glasses state. We consider that the optical correction effect of the refractive error has resulted from the stabilization of the activity of the somatic system among the interactions of the sensory organs responsible for postural control. These findings may have important clinical implications in the field of optometry and balance evaluation. In particular, cooperation between specialists in each field is required for elderly or low vision patients with limited mobility and poor quality of life.

## Supporting information

**S1 Table. All relevant raw data.**
(XLSX)

## Author Contributions

**Conceptualization:** Ji In Bae, Dong-Sik Yu, Sang-Yeob Kim.

**Data curation:** Ji In Bae, Sang-Yeob Kim.

**Formal analysis:** Ji In Bae, Dong-Sik Yu, Sang-Yeob Kim.

**Methodology:** Ji In Bae, Sang-Yeob Kim.

**Project administration:** Sang-Yeob Kim.

**Resources:** Ji In Bae.

**Supervision:** Sang-Yeob Kim.

**Validation:** Ji In Bae, Dong-Sik Yu, Sang-Yeob Kim.

**Visualization:** Dong-Sik Yu, Sang-Yeob Kim.

**Writing – original draft:** Ji In Bae.

**Writing – review & editing:** Dong-Sik Yu, Sang-Yeob Kim.

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
