## [Decision Letter · Decision Letter 0]

14 Apr 2020

PONE-D-20-06787

The effect of optical correction by new-glasses for refractive errors on postural stability

PLOS ONE

Dear Mr. Kim,

Thank you for submitting your manuscript to PLOS ONE. After careful consideration, we feel that it has merit but does not fully meet PLOS ONE’s publication criteria as it currently stands. Therefore, we invite you to submit a revised version of the manuscript that addresses the points raised during the review process.

Please refer to my comments below in framing your revisions.

We would appreciate receiving your revised manuscript by May 29 2020 11:59PM. To enhance the reproducibility of your results, we recommend that if applicable you deposit your laboratory protocols in protocols.io, where a protocol can be assigned its own identifier (DOI) such that it can be cited independently in the future. For instructions see: http://journals.plos.org/plosone/s/submission-guidelines#loc-laboratory-protocols

We look forward to receiving your revised manuscript.

Kind regards,

Thomas A Stoffregen, PhD

Academic Editor

PLOS ONE

Journal Requirements:

1. We note that Figure [1] includes an image of a [patient / participant / in the study]. 

Additional Editor Comments (if provided):

I have received reviews from two experts on postural control. Please use their comments in revising your manuscript. Note especially the design issue raised by Reviewer 1, and the issue (raised by both Reviewers) relating to your use of inferential statistics. I agree with the Reviewers that t-tests may be i appropriate for many of the comparisons that you have reported. With respect to the design issue raised by Reviewer 1, please refer to this explicitly in the Cover Letter summarizing changes you have made, if you elect to submit a revision the PLOS ONE.

Reviewers' comments:

Reviewer's Responses to Questions

**Comments to the Author**

1. Is the manuscript technically sound, and do the data support the conclusions?

Reviewer #1: No

Reviewer #2: Partly

2. Has the statistical analysis been performed appropriately and rigorously? 

Reviewer #1: No

Reviewer #2: No

3. Have the authors made all data underlying the findings in their manuscript fully available?

Reviewer #1: Yes

Reviewer #2: Yes

4. Is the manuscript presented in an intelligible fashion and written in standard English?

Reviewer #1: Yes

Reviewer #2: No

5. Review Comments to the Author

Reviewer #1: 1. Line 21: New glasses can be interpreted as either first-time wearing glasses, or as updated prescription. This issue is repeated throughout the manuscript.

2. Introduction appears to be well-research and strongly motivated.

3. Line 87: What is “real” myopia?

4. Subjects

There seems to be a significant confusion here in terms of experimental design.

The Myopia and Hyperopia groups are receiving two different conditions.

a. Hyperopia is receiving glasses for the first time: a novel experience

b. Myopia is receiving updates lenses: a non-novel experience

While any significant change in prescription will have an adjustment period, it is important to investigate the postural differences between an updated prescription and the entirely new experience of getting glasses. Until there is data stating that the differences between these two conditions are statistically non-significant, you cannot proceed on the assumption that they are comparable. That is to say, the outcomes of these two groups cannot be compared, because they are essentially receiving different treatments (first time glasses vs updated lenses).

I do commend the researchers for ensuring that the average diopter change to the lenses was comparable between the myopia and hyperopia group. With this being stated, in order to publish the data in this study, these two groups should be separated into either two separate papers or into two distinctively different sections of the paper.

c. Effects of updated prescriptions in patients with Myopia on postural sway.

d. Effects of first-time glasses worn by patients with Hyperopia on postural sway.

5. Line 119: “posture”

6. Line 168: A paired t-test is appropriate for two sets of observations from a same subject. This is done throughout the manuscript, but a two-way ANOVA is most appropriate for the conclusions being drawn here. However, this is still not sufficient, because of the two different starting conditions.

Discussion:

7. Line 250: “Man” should be “Many”

While this manuscript is well-motivated, the immediate issue of the experimental design is glaring. There may be significant differences between myopia and hyperopia when receiving new glasses, as well as when receiving an updated prescription. There is not enough collected data to properly complete either of those comparisons.

Ideally, a repeated measures ANOVA would be the best statistical test for your end goal. (Myopia x Hyperopia) x (Without x Old x New) While the data in this manuscript may be usable, the emphasis between these two very different starting conditions needs to be stressed.

At this point, major revision is advisable at a minimum. The manuscript will need to be divided into two sections, or redone as two separate papers.

Reviewer #2: In this study, standing postural motion was recorded and analyzed before and after participants were provided with glasses that corrected their vision to normal. The authors where interested in the influence of the quality of visual input (impaired/corrected) on postural sway indices. Authors found a reduction in postural instability with the introduction of corrective lenses (the effect was greater for myopic vs. hyperopic participants). This was a straightforward empirical work with direct clinical/applied applications.

Format issues:

Need to have someone proofread for English (phrasing, tense, word choice) throughout document - the translation to English is not entirely correct. Also need to check for typographical and spelling errors throughout the document. In particular avoid the term "prove(d)" your research can only 'demonstrate' or 'support'.

Table 1 should be broken into smaller subsections for ease of reading/comprehension (a table spanning two-pages is hard to follow).

What (if any) are the units for your dependent measures - this should be stated in the method/results and in the data figures/tables as it is hard to infer significance of the differences without a reference point (metric)

Analysis issues:

In many places it seems like you are performing multiple t-tests on the same data- I would caution against this as it will increase the likelihood of producing false positives. In particular, the data organization of Tables 2 -4 suggest an analysis of variance (ANOVA) would have been more appropriate.

Method question:

given that the visual target was 6 m distant from the participants wouldn't that advantage one type of visual correction over the other?

6. PLOS authors have the option to publish the peer review history of their article (what does this mean?). If published, this will include your full peer review and any attached files.

Reviewer #1: No

Reviewer #2: Yes: L. James Smart Jr.

---

## [Author Response · Author response to Decision Letter 0]

21 May 2020

Manuscript number: PONE-D-20-06787

Title: The effect of optical correction by new-glasses for refractive errors on postural stability 

Journal Requirements (Marked as J): 

J1. We note that Figure [1] includes an image of a [patient / participant / in the study]. 

As per the PLOS ONE policy (http://journals.plos.org/plosone/s/submission-guidelines#loc-human-subjects-research) on papers that include identifying, or potentially identifying, information, the individual(s) or parent(s)/guardian(s) must be informed of the terms of the PLOS open-access (CC-BY) license and provide specific permission for publication of these details under the terms of this license. Please download the Consent Form for Publication in a PLOS Journal (http://journals.plos.org/plosone/s/file?id=8ce6/plos-consent-form-english.pdf). The signed consent form should not be submitted with the manuscript, but should be securely filed in the individual's case notes. Please amend the methods section and ethics statement of the manuscript to explicitly state that the patient/participant has provided consent for publication: “The individual in this manuscript has given written informed consent (as outlined in PLOS consent form) to publish these case details”. If you are unable to obtain consent from the subject of the photograph, you will need to remove the figure and any other textual identifying information or case descriptions for this individual.

Answer for [J1]: We have changed Fig.1 (without the subject).

Additional Editor Comments 

I have received reviews from two experts on postural control. Please use their comments in revising your manuscript. Note especially the design issue raised by Reviewer 1, and the issue (raised by both Reviewers) relating to your use of inferential statistics. I agree with the Reviewers that t-tests may be i appropriate for many of the comparisons that you have reported. With respect to the design issue raised by Reviewer 1, please refer to this explicitly in the Cover Letter summarizing changes you have made, if you elect to submit a revision the PLOS ONE. 

Response of authors: We are so grateful to the editor for reviewing our manuscript PONE-D-20-06787, titled “The Effect of Optical Correction by New-Glasses for Refractive Errors on Postural Stability,” submitted for publication in PLOS ONE. We also thank the reviewers for their detailed comments. 

Refractive errors may not be a serious issue because they do not cause fatal problems compared to other eye diseases. However, the purpose of this study was to highlight that clinical implications of a study in which the vision and postural controls converged are essential in the areas of balance evaluation and optometry. Additionally, the role of optometrist managing primary eye care is crucial in modern society wherein communication and coordination between studies are required. 

Although it can be classified into first-time glasses and updated glasses depending on the experience of wearing glasses, the updated glasses for myopes and first time wearing glasses for hyperopes can be considered as full corrected glasses in each type of refractive error. To reflect this, a title has changed from “The Effect of Optical Correction by New-Glasses for Refractive Errors on Postural Stability” to “Effect of Optical Correction by Fully Corrected Glasses on Postural Stability”. Also, myopes and hyperopes were completely separated and analyzed in the revised manuscript, respectively, to resolve the experimental design and statistical analysis issues pointed out by the reviewers. 

We’ve done our best to provide reasonable answers to address issues and questions raised by the reviewers. Also, this manuscript for resubmission has been edited by a native English speaker according to the reviewer’s recommendations. Enclosed, please find the revised manuscript with the changes highlighted in colored text. 

Reviewers’ Comments ; Reviewer’s Responses to Questions

Comments to the Author

1. Is the manuscript technically sound, and do the data support the conclusions?

Reviewer #1: No

Reviewer #2: Partly________________________________________

2. Has the statistical analysis been performed appropriately and rigorously? 

Reviewer #1: No

Reviewer #2: No

3. Have the authors made all data underlying the findings in their manuscript fully available?

The PLOS Data Policy requires authors to make all data underlying the findings described in their manuscript fully available without restriction, with rare exception (please refer to the Data Availability Statement in the manuscript PDF file). The data should be provided as part of the manuscript or its supporting information, or deposited to a public repository. For example, in addition to summary statistics, the data points behind means, medians and variance measures should be available. If there are restrictions on publicly sharing data—e.g. participant privacy or use of data from a third party—those must be specified.

Reviewer #1: Yes

Reviewer #2: Yes

4. Is the manuscript presented in an intelligible fashion and written in standard English?

Reviewer #1: Yes

Reviewer #2: No 

5. Review Comments to the Author

※ Reviewer #1 (Marked as R1): 

R1-1. Line 21: New glasses can be interpreted as either first-time wearing glasses, or as updated prescription. This issue is repeated throughout the manuscript.

Answer for [R1-1]: We thank the reviewer for the detailed comments. As explained below, myopes and hyperopes were completely separated and analyzed in the revised manuscript, respectively, according to your advice. Although it can be classified into first-time glasses and updated glasses depending on the experience of wearing glasses, the updated glasses for myopes and first-time wearing glasses for hyperopes can be considered as full corrected glasses in each type of refractive error. The authors consider that it more appropriate to use the term “full corrected glasses” rather than new glasses. So, in the revised manuscript, we have revised the term ‘new glasses’ used in submitted paper to ‘full corrected glasses as the main key word’. (Please see lines 22, 29, 34, 115, 121, 185, 188, 337 and 431 in tracked changes manuscript) 

R1-2. Introduction appears to be well-research and strongly motivated.

Answer for [R1-2]: We appreciate your positive evaluation.

R1-3. Line 87: What is “real” myopia?

Answer for [R1-3]: We are sorry about the unclear description. Real myopia was an expression to distinguish it from experimentally induced myopia. But, we agree there may be some confusion here. So, the word “real” was deleted to avoid confusion. (Please see line 102)

R1-4. Subjects

There seems to be a significant confusion here in terms of experimental design.

The Myopia and Hyperopia groups are receiving two different conditions.

a. Hyperopia is receiving glasses for the first time: a novel experience

b. Myopia is receiving updates lenses: a non-novel experience

While any significant change in prescription will have an adjustment period, it is important to investigate the postural differences between an updated prescription and the entirely new experience of getting glasses. Until there is data stating that the differences between these two conditions are statistically non-significant, you cannot proceed on the assumption that they are comparable. That is to say, the outcomes of these two groups cannot be compared, because they are essentially receiving different treatments (first time glasses vs updated lenses).

I do commend the researchers for ensuring that the average diopter change to the lenses was comparable between the myopia and hyperopia group. With this being stated, in order to publish the data in this study, these two groups should be separated into either two separate papers or into two distinctively different sections of the paper.

c. Effects of updated prescriptions in patients with Myopia on postural sway.

d. Effects of first-time glasses worn by patients with Hyperopia on postural sway.

Answer for [R1-4]: We thank the reviewer for the positive critique. The reason why we did not distinguish the experience of wearing glasses in these two refractive errors type was that the purpose of this study was to emphasize the effect of optically full corrected glasses regardless of the type of refractive errors. Since the visual condition differs significantly according to age, we conducted this study under the condition of the same age range for the subjects. However, it was very difficult to recruit young subjects already wearing glasses for correcting hyperopia, because they enable focus of images to the level attained as emmetropia by automatic focusing (accommodation) without optical correction. 

As the reviewer pointed out, we acknowledge that this study could not fully explain the adaption issue depending on their glasses wearing experience (first-time glasses vs. updated glasses). Although our results show the difficulty in establishing the adaptation phenomenon of wearing glasses, we think that the relationship between the postural stability and their glasses wearing experience may be a key clue in resolving this issue. So, based on your advice, we will investigate the effect of adaptation phenomenon when wearing glasses in the next experiment. Your advice will lead to our next study and we really appreciate it. So, based on your advice, myopes and hyperopes were completely separated and analyzed in the revised manuscript, respectively. 

The main changes are as follow: 

<Title>

- A title has changed from “The Effect of Optical Correction by New-Glasses for Refractive Errors on Postural Stability” to “Effect of Optical Correction by Fully Corrected Glasses on Postural Stability”. (Please see line 1)

<Subjects>

- Two subjects with no experience in wearing glasses among 19 myopes were excluded to satisfy a condition for updated glasses in the revised version. The changed texts are as follows: (Please see line 126-131)

“This study included 34 subjects (19 males and 15 females) of a mean age of 22.39 ± 2.30 years-old. Among the 34 subjects, 17 subjects, of an average age of 21.18±1.59 years, had myopic refractive errors, including simple myopia and myopic astigmatism, and 17 subjects of an average age of 23.59±2.53 years had hyperopic refractive errors, including simple hyperopia and hyperopic astigmatism. Among all subjects, 17 myopes were already using prescription glasses (old-glasses) and 17 hyperopes were not using prescription glasses”.

<Data analysis>

- A repeated-measures ANOVA was newly performed for 17 myopes (no glasses vs, old glasses vs, updated glasses) and the paired t-test was performed for 17 hyperopes (no glasses vs, first-time glasses) in the revised version. (Please see revised Fig 3 and Fig 4, Table 3)

- We rewrote the results of the “Abstract section” based on the changed results. The changed texts are as follows: 

(Please see lines 25-29) “To this end, a total of 34 subjects (19 males and 15 females) of an average age of 22.38 ± 2.41-years-old participated in this study. After measuring the full corrected powers of refractive errors of subjects through subjective refraction, updated glasses were provided to 17 myopic subjects and first-time glasses were provided to 17 hyperopic subjects as full corrected glasses, respectively.”

- We rewrote the “Results section” based on the changed results. The changed texts are as follow: 

(Please see lines 210-215) “Table 1 summarizes the glasses prescriptions for 17 myopic subjects. The average equivalent spherical power of their old glasses was S-3.92 ± 1.82 D, and the average equivalent spherical power of full corrected prescriptions for updated glasses was S-4.11 ± 1.92 D. Table 2 presents the full corrected prescriptions in hyperopes. In 17 hyperopic subjects, the average equivalent spherical power of full corrected prescriptions for their first-time glasses was S+0.43 ± 0.29 D.”

(Please see lines 236-249) “After wearing the corrective glasses in the 17 myopes, the average change in postural instability index is shown in Fig 3. The postural instability index was significantly decreased when wearing myopia correction glasses as compared to the value measured in the no-glasses state (F = 4.561, p = 0.020). A post-hoc analysis revealed that the postural instability index was significantly decreased in the old glasses state and after the wearing the updated glasses compared to the values measured in the no glasses state (p = 0.040 for no glasses vs. old glasses, p = 0.018 for no glasses vs. updated glasses). However, compared to the old glasses, the postural instability tended to decrease after wearing the updated glasses, but there was no statistically significant difference (p = 0.544 for old glasses vs. updated glasses).

Fig. 4 shows the analysis results of comparing the average postural instability of the 17 hyperopes with first time glasses. After wearing hyperopia corrective glasses, the postural instability index was more significantly decreased than the no glasses state, but there was no statistically significant difference (t = 1.006, p = 0.329).”

(Please see lines 282-295) “Table 3 shows the results of having analyzed and compared the sway power index in the four frequency regions after wearing the corrective glasses in 17 myopic subjects. A significant difference in sway power index was only observed in the mid-high frequency region among 4 frequency regions, depending on visual conditions (F = 4.724, p = 0.021). A post-hoc analysis revealed that the sway power index in the mid-high frequency was significantly decreased in the old glasses state and after wearing the updated glasses compared to the values measured in the no glasses state (p = 0.006 for no glasses vs. old glasses, p = 0.016 for no glasses vs. updated glasses). However, compared to the old glasses, postural instability tended to decrease after wearing the updated glasses, but there was no statistically significant difference (p = 0.830 for old glasses vs. updated glasses). The changes in the sway power index in each frequency region before and after wearing the first time glasses in 17 hyperopic subjects are shown in Table 4. Contrary to myopes, there was no significant difference in sway power index in all frequency regions even though fully corrected glasses were worn.”

<Discussion> 

- Discussion was divided into three sections as follow: 

1. Effects of full optical correction for myopes on postural stability (Please see lines 334-374)

2. Effects of full optical correction for hyperopes on postural stability (Please see lines 377-394) 

3. Fourier transformation analysis of sway power index before and after wearing corrective glasses (Please see lines 396-434)

- The limitations of this study were added as follows:(Please see lines 429-434) 

“However, the results of this study are limited since they were based on the results collected over the course of 32 seconds after wearing the full corrected glasses, whether in the form of updated or first-time glasses. Therefore, further research should be conducted to further analyze the adaptation phenomenon when wearing glasses, as well as the time-dependent changes following optical correction.” 

- It was added at the end of the pater for clinical implications as follows: (Please see lines 446-447) 

“In particular, cooperation between specialists in each field is required for elderly or low vision patients with limited mobility and poor quality of life.”

R1-5. Line 119: “posture”

Answer for [R1-5]: Thank you for your detailed comments. We have changed “poture” to “posture”. (Please see line 149)

R1-6. Line 168: A paired t-test is appropriate for two sets of observations from a same subject. This is done throughout the manuscript, but a two-way ANOVA is most appropriate for the conclusions being drawn here. However, this is still not sufficient, because of the two different starting conditions.

Answer for [R1-6]: As the reviewer pointed out, statistical analysis was performed again by dividing it into two sections. A repeated-measures ANOVA was performed for 17 myopes (no glasses vs, old glasses vs, updated glasses) and paired t-test was performed for 17 hyperopes (no glasses vs, first time glasses) in the revised version. And we have revised the manuscript as follow: 

(Please see lines 202-206) “The paired t-test and repeated-measures analysis of variance (repeated-measures ANOVA) methods were used to analyze the changes in postural control in each measurement condition. For all analyses, it was estimated that there was a statistically significant difference when p < 0.05.”

Discussion:

R1-7. Line 250: “Man” should be “Many”

Answer for [R1-7]: Thank you for your detailed comments. We have changed “Man” to “Many”. (Please see line 335)

R1-8. While this manuscript is well-motivated, the immediate issue of the experimental design is glaring. There may be significant differences between myopia and hyperopia when receiving new glasses, as well as when receiving an updated prescription. There is not enough collected data to properly complete either of those comparisons.

Ideally, a repeated measures ANOVA would be the best statistical test for your end goal. (Myopia x Hyperopia) x (Without x Old x New) While the data in this manuscript may be usable, the emphasis between these two very different starting conditions needs to be stressed.

At this point, major revision is advisable at a minimum. The manuscript will need to be divided into two sections, or redone as two separate papers.

Answer for [R1-8]: According to your advice, statistical analysis was performed again by dividing it into two sections. A repeated-measures ANOVA was performed for 17 myopes (no glasses vs, old glasses vs, updated glasses) and the paired t-test was performed for 17 hyperopes (no glasses vs, first-time glasses) in the revised version. Based on the changed results, Fig. 3 and Fig. 4, and Table 3 were revised.

※ Reviewer #2 (Marked as R2): 

R2-1. In this study, standing postural motion was recorded and analyzed before and after participants were provided with glasses that corrected their vision to normal. The authors where interested in the influence of the quality of visual input (impaired/corrected) on postural sway indices. Authors found a reduction in postural instability with the introduction of corrective lenses (the effect was greater for myopic vs. hyperopic participants). This was a straightforward empirical work with direct clinical/applied applications.

Answer for [R2-1]: In our previous study, we compared the minimum refractive powers that affect postural stability for each type of refractive errors by inducing various levels of myopic and hyperopic power through (±) spherical lenses. However, results of the previous study have limited validity and were restricted to experimentally induced refractive errors. Even though design of this study is very simple and straightforward empirical work as you mention, we think this study is absolutely necessary to provide a more realistic validation in myopes and hyperopes.

R2-2. Format issues:

Need to have someone proofread for English (phrasing, tense, word choice) throughout document - the translation to English is not entirely correct. Also need to check for typographical and spelling errors throughout the document. In particular avoid the term "prove(d)" your research can only 'demonstrate' or 'support'.

Answer for [R2-2]: Our paper underwent language editing before submission. And, a second language editing was also performed after the revision process. We uploaded the confirmation documents for the language editing. Also, according to your advice, we have changed the term of “prove” to “demonstrate”. (Please see line 51, 114)

R2-3. Table 1 should be broken into smaller subsections for ease of reading/comprehension (a table spanning two-pages is hard to follow).

Answer for [R2-3]: According to your advice, Table 1 is divided into Table 1 and 2. (Please see line 228, 232)

R2-4. What (if any) are the units for your dependent measures - this should be stated in the method/results and in the data figures/tables as it is hard to infer significance of the differences without a reference point (metric)

Answer for [R2-4]: Thank you for your comments. A Tetrax device comprises four independent mobile force plates, and these plates are positioned to measure any equilibrium disturbance from two forefeet and two heels. By taking the average of all four measurements, it provides a description of body sway in terms of displacement of the patient’s center of pressure. It is possible to analyze general stability and Fourier spectral analysis, and these are calculated as an index [Ref.]. In the case of postural instability, the unit “index” is already indicated in all texts, but, the unit notation for sway power is missing in texts. Thus, we have revised “sway power” to “sway power index” all texts and in Table 3-Table 4 in to resolve the unit issue. (Please see line 34, 44, 48, 56, 120, 153, 169, 197, 282, 284, 286, 292, 294, 314, 328, 396, 399, 402, 410, 411, and 417)

[Ref.] Alpini D, Berardino FD, Mattei V, et al. Characteristics of Multiple Sclerosis Patient Stance Control Disorders, Measrued by Means of Posturography and Related to Brainstem Lesions. Audiol Res 2012;2:43-9.

R2-5. Analysis issues:

In many places it seems like you are performing multiple t-tests on the same data- I would caution against this as it will increase the likelihood of producing false positives. In particular, the data organization of Tables 2-4 suggest an analysis of variance (ANOVA) would have been more appropriate.

Answer for [R2-5]: Thank you for your suggestion. Table 2 in the original manuscript was deleted due to an experimental design problem pointed out by another reviewer. In the revised manuscript, a repeated-measures ANOVA was newly performed for 17 myopes (no glasses vs, old glasses vs, updated glasses, revised Table 3). But, Table 4 (no glasses vs, first-time glasses in hyperopes) was not revised because the statistical analysis of Table 3 and Table 4 is for comparison before and after wearing glasses in four frequency ranges. Table 3 and Fig. 3 and Fig. 4 were revised based on the changed results.

R2-6. Method question:

given that the visual target was 6 m distant from the participants wouldn't that advantage one type of visual correction over the other?

Answer for [R2-6]: According to the literature[Ref.], the recommended distance for visual acuity test is 6 m. An added complication in the determination of the final spherical endpoint rests in the fact that most test distances are seldom in excess of 6 m and can be as minimal as 4 m. It is frequently overlooked that the distance of 6 m actually represents a fixation point effectively equal to +0.16 DS (conversion as Diopter: 1/6 m = 0.16 D) of accommodation. Hence, eyes are often under-minused (in case of myopia) or over-plussed (in case of hyperopia) by a slight amount. However, this value is only arithmetic, and the minimum unit for prescription lenses is 0.25 DS. So, issue of distance is clinically negligible, if the test is performed at 6 m.

[Ref.]. Benjamin WJ. Borish’s Clinical Refraction. 2nd ed. Oxford, UK: Butterworth-Heinemann; 2006. Chapter 20. 835p ________________________________________

6. PLOS authors have the option to publish the peer review history of their article (what does this mean?). If published, this will include your full peer review and any attached files.

Do you want your identity to be public for this peer review? For information about this choice, including consent withdrawal, please see our Privacy Policy.

Reviewer #1: No

Reviewer #2: Yes: L. James Smart Jr.

< The End Reply>

---

## [Decision Letter · Decision Letter 1]

11 Jun 2020

PONE-D-20-06787R1

Effect of optical correction by fully corrected glasses on postural stability

PLOS ONE

Dear Dr. Kim,

Thank you for submitting your manuscript to PLOS ONE. After careful consideration, we feel that it has merit but does not fully meet PLOS ONE’s publication criteria as it currently stands. Therefore, we invite you to submit a revised version of the manuscript that addresses the points raised during the review process.

As you will see, both Reviewers approved your revisions, and request only a few modest changes. I endorse their recommendations. I look forward to accepting a modestly revised final version.

We look forward to receiving your revised manuscript.

Kind regards,

Thomas A Stoffregen, PhD

Academic Editor

PLOS ONE

Reviewers' comments:

Reviewer's Responses to Questions

**Comments to the Author**

1. If the authors have adequately addressed your comments raised in a previous round of review and you feel that this manuscript is now acceptable for publication, you may indicate that here to bypass the “Comments to the Author” section, enter your conflict of interest statement in the “Confidential to Editor” section, and submit your "Accept" recommendation.

Reviewer #1: All comments have been addressed

Reviewer #2: (No Response)

2. Is the manuscript technically sound, and do the data support the conclusions?

Reviewer #1: Yes

Reviewer #2: Yes

3. Has the statistical analysis been performed appropriately and rigorously? 

Reviewer #1: Yes

Reviewer #2: Yes

4. Have the authors made all data underlying the findings in their manuscript fully available?

Reviewer #1: Yes

Reviewer #2: (No Response)

5. Is the manuscript presented in an intelligible fashion and written in standard English?

Reviewer #1: Yes

Reviewer #2: Yes

6. Review Comments to the Author

Reviewer #1: Line 198: Add clarity here, by stating which analysis was used. Repeated measures ANOVA, due to the presence of F. Same should go for any analysis (t-test, post-hoc, ANOVAs, etc).

Line 209: “instability index was more significantly decreased”

You do state afterwards that this is not statistically significant. I would remove the “significantly” in line 209 to bolster clarity.

Review indenting/consistency in formatting, overall.

I greatly appreciate your response, and wish you the best for future research along this path.

Reviewer #2: This manuscript is much improved - thank you for addressing previous concerns.

there are a couple of minor comments that should be addressed:

Intro

Line 75-76 “The findings of these studies suggest the importance of examining the impact of visual information on postural control” – suggested phrasing.

Line 81, 84 – remove “the” before postural instability

Line 87 – should this be “25% more body sway” instead of “25% of body sway”?

Results:

The postural instability index – appears to be a scalar (metric free) measure – is that correct? If it doesn’t have a metric – then you might consider providing a range of values so that the reader has a point of reference.

Unless the journal requires exact values for reporting significance just use P < .05 or P > .05 as that was your criterion. Exact p values often lead people to think of an effect as more or less significant). If you wish to convey the practical significance – you can report effect sizes.

7. PLOS authors have the option to publish the peer review history of their article (what does this mean?). If published, this will include your full peer review and any attached files.

Reviewer #1: No

Reviewer #2: Yes: L. James Smart Jr.

---

## [Author Response · Author response to Decision Letter 1]

23 Jun 2020

Manuscript number: PONE-D-20-06787R1

Title: Effect of optical correction by fully corrected glasses on postural stability

Editor Comments 

Dear Dr. Kim,

Thank you for submitting your manuscript to PLOS ONE. After careful consideration, we feel that it has merit but does not fully meet PLOS ONE’s publication criteria as it currently stands. Therefore, we invite you to submit a revised version of the manuscript that addresses the points raised during the review process.

As you will see, both Reviewers approved your revisions, and request only a few modest changes. I endorse their recommendations. I look forward to accepting a modestly revised final version.

Review Comments to the Author 

※ Reviewer #1 (Marked as R1): 

R1-1. Line 198: Add clarity here, by stating which analysis was used. Repeated measures ANOVA, due to the presence of F. Same should go for any analysis (t-test, post-hoc, ANOVAs, etc).

Response to [R1-1]: Thank you for your detailed comments. As the reviewer pointed out, each statistical method was added to the Results section as follows. (Please see the lines in the “Revised Manuscript with Track Changes”)

Lines 203-204: “(F = 4.561, p < 0.05 by repeated measures ANOVA)”

Lines 206-208: “(p < 0.05 for no glasses vs. old glasses, p < 0.05 for no glasses vs. updated glasses by LSD post hoc analysis)”

Line 210: “(p > 0.05 for old glasses vs. updated glasses by LSD post hoc analysis)”

Lines 214-215: “(t = 1.006, p > 0.05 by paired t-test)”

Lines 230-231: “(F = 4.724, p < 0.05 by repeated measures ANOVA)”

Lines 234-235: “(p < 0.05 for no glasses vs. old glasses, p < 0.05 for no glasses vs. updated glasses by LSD post hoc analysis)”

Line 237-238: “(p > 0.05 for old glasses vs. updated glasses by LSD post hoc analysis)”

For further clarification, footnotes were added to Figures 3 and 4 as follows.

The footnote in Fig 3 (Please see lines 219-221): “*p < 0.05: significantly different depending on each visual condition according to LSD (least significant difference) post hoc analysis by repeated measures ANOVA” 

Error bars indicate the standard error (SE) of the mean.

The footnote in Fig 4 (Please see line 225): “Error bars indicate the standard error (SE) of the mean” 

R1-2. Line 209: “instability index was more significantly decreased”

You do state afterwards that this is not statistically significant. I would remove the “significantly” in line 209 to bolster clarity.

Response to [R1-2]: Thank you for your comments. I agree with the reviewers. “Significantly” was removed, as you suggested (Please see line 213).

R1-3. Review indenting/consistency in formatting, overall. I greatly appreciate your response, and wish you the best for future research along this path.

Response to [R1-3]: Thank you for your comments. We revised the format of the main body (Paragraph

Indentation, References et.al). We are very grateful to the reviewer for reviewing our manuscript. 

※ Reviewer #2 (Marked as R2): 

This manuscript is much improved - thank you for addressing previous concerns.

there are a couple of minor comments that should be addressed:

Intro

R2-1. Line 75-76 “The findings of these studies suggest the importance of examining the impact of visual information on postural control” – suggested phrasing.

Response to [R2-1]: Thank you for your good suggestion. We revised the manuscript using your suggested phrasing (Please see lines 74-76).

R2-2. Line 81, 84 – remove “the” before postural instability

Response to [R2-2]: Thank you for your comments. “The” was removed, as your suggested (Please see lines 82 and 85). 

R2-3. Line 87 – should this be “25% more body sway” instead of “25% of body sway”?

Response to [R2-3]: Thank you for your detailed comments. We revised the text from “25% of body sway” to “25% more body sway” according to your advice (Please see line 88).

Results:

R2-4. The postural instability index – appears to be a scalar (metric free) measure – is that correct? If it doesn’t have a metric – then you might consider providing a range of values so that the reader has a point of reference.

Response to [R2-4]: Thank you for your detailed comments. As you pointed out, the postural instability index is a scalar (metric free) measure, so the range of values was added as a point of reference for the reader as follows. (Please see lines 138-140).

“The postural instability index also increases with age as a result of degraded postural control (11.69±2.21 in normal young subjects and 24.84±6.07 in older subjects over 65 years old) [26].”

 R2-5. Unless the journal requires exact values for reporting significance just use P < .05 or P > .05 as that was your criterion. Exact p values often lead people to think of an effect as more or less significant). If you wish to convey the practical significance – you can report effect sizes. 

Response to [R2-5]: Thank you for your detailed comments. Based on your advice, we revised the p-values in Results section of main text as p < 0.05 or p > 0.05, but the p-values in Tables do not change to show exact effect sizes for reader (Please see the lines 203, 206, 207, 210, 214, 231, 234, and 237)

 < End of the response>

---

## [Editor Report · Decision Letter 2]

25 Jun 2020

Effect of optical correction by fully corrected glasses on postural stability

PONE-D-20-06787R2

Dear Dr. Kim,

We’re pleased to inform you that your manuscript has been judged scientifically suitable for publication and will be formally accepted for publication once it meets all outstanding technical requirements.

Kind regards,

Thomas A Stoffregen, PhD

Academic Editor

PLOS ONE
---

## [Editor Report · Acceptance letter]

29 Jun 2020

PONE-D-20-06787R2 

Effect of optical correction by fully corrected glasses on postural stability 

Dear Dr. Kim:

I'm pleased to inform you that your manuscript has been deemed suitable for publication in PLOS ONE. Congratulations! Your manuscript is now with our production department. 

Kind regards, 

on behalf of

Dr. Thomas A Stoffregen 

Academic Editor

PLOS ONE